# Investigations on the Influence of the Axial Ligand in [Salophene]iron(III) Complexes on Biological Activity and Redox Behavior

**DOI:** 10.3390/ijms24032173

**Published:** 2023-01-21

**Authors:** Hubert Descher, Sophie Luise Strich, Martin Hermann, Peter Enoh, Brigitte Kircher, Ronald Gust

**Affiliations:** 1Department of Pharmaceutical Chemistry, Institute of Pharmacy, CMBI—Center for Molecular Biosciences Innsbruck, CCB—Center for Chemistry and Biomedicine, University of Innsbruck, Innrain 80-82, 6020 Innsbruck, Austria; 2Tyrolean Cancer Research Institute, Innrain 66, 6020 Innsbruck, Austria; 3Immunobiology and Stem Cell Laboratory, Department of Internal Medicine V (Hematology and Oncology), Innsbruck Medical University, Anichstraße 35, 6020 Innsbruck, Austria; 4Department of Anesthesiology and Critical Care Medicine, Innsbruck Medical University, Anichstraße 35, 6020 Innsbruck, Austria

**Keywords:** iron(III) salophene, ferroptosis, apoptosis, necroptosis, cytotoxicity, cyclic voltammetry, redox potential, ROS

## Abstract

The [*N,N′*-disalicylidene-1,2-phenylenediamine]iron(III) ([salophene]iron(III)) derivatives **1**–**4** with anionic axial ligands (A = Cl^−^, NO_3_^−^, SCN^−^, CH_3_COO^−^) and complexes **5** and **6** with neutral ligands (A = imidazole, 1-methylimidazole) as well as the μ-oxo dimer **7** inhibited proliferation, reduced metabolic activity, and increased mitochondrial reactive oxygen species. Ferroptosis as part of the mode of action was identified by inhibitor experiments, together with induction of lipid peroxidation and diminished mitochondrial membrane potential. No differences in activity were observed for all compounds except **4**, which was slightly less active. Electrochemical analyses revealed for all compounds a fast attachment of the solvent dimethyl sulfoxide and a release of the axial ligand A. In contrast, in dichloromethane and acetonitrile, ligand exchange did not take place, as analyzed by measurements of the standard potential for the iron(III/II) redox reaction.

## 1. Introduction

[Salophene]iron(III) complexes (salophene = *N,N′*-disalicylidene-1,2-phenylenediamine) have already been extensively studied regarding their catalytic activity [1] and their cytotoxic properties [2,3,4,5,6], whereby the chlorido[salophene]iron(III) complex **1** has been identified as the most potent compound of these series [5,7,8]. Its antitumor activity is mainly attributed to the induction of mitochondrial reactive oxygen species (mROS) and ferroptosis [5]. The complete mechanism of action behind ferroptosis has not been elucidated yet. However, it has been shown that iron-mediated ROS production by the Fenton reaction promotes lipid peroxidation and oxidation of polyunsaturated fatty acids of membrane lipids, resulting in membrane damage [9,10]. Complex **1** caused decomposition of hydrogen peroxide similar to the Fenton reaction and supports the hypothesis that not only free iron(II) ions initiate the formation of lipid ROS and ferroptosis [5,11]. The electrochemistry of salophene complexes has been previously described [12,13,14,15]. Carré and colleagues analyzed **1** and its μ-oxo dimer **7** in acetonitrile (ACN) and dichloromethane (DCM) [13,14]. In comparison to free iron(III) ions, the chelated iron(III) species showed a lower tendency toward reduction due to a higher electron density caused by the ligands, which donate electrons to the iron(III) center. The oxygen bridge in **7** shifted the signal to a more anodic potential and documented strong electron donating properties [14].

Structural modification at the salicylidene [2,3,7,16,17] and the phenylenediamine moiety [3,5,6,7] affects the biological activity of the complexes. In addition, the influence of the axially coordinated ligand on the antiproliferative effects was evaluated [4]. However, a clear determination of possible exchange reactions and the influence on the redox behavior of the complexes are missing. Therefore, in continuation of our studies on the antitumor effects of [salophene]iron(III) complexes, the axially bound chlorido ligand (A) in **1** was exchanged by various anionic (A = NO_3_^−^, SCN^−^, CH_3_COO^−^) and neutral (A = imidazole, 1-methylimidazole) groups (Figure 1), which were selected based on the hard and soft acids and bases (HSAB) concept, which characterizes iron(III) as a hard acid [18]. The ligand A acts either as an electron donor or acceptor, which should influence, on the one hand, its binding strength to the metal and, on the other hand, the redox potential of the complex. Therefore, complexes **1**–**7** were examined for their electrochemical behavior via cyclic voltammetry in different solvents, including DCM, ACN, dimethylformamide (DMF), and dimethyl sulfoxide (DMSO), and for biological activity in the Cisplatin-resistant ovarian carcinoma A2780cis and mammary carcinoma MDA-MB-231 cell lines. 

## 2. Results and Discussion

### 2.1. Synthetic Chemistry

#### 2.1.1. Synthesis of the Compounds

The synthesis of complexes **1**–**7** is depicted in Figure 2. According to a previously published method, the salophene ligand was produced from salicylic aldehyde and 1,2-phenylenediamine [19]. Coordination reaction of the ligand with the respective iron(III) salt (method a: FeCl_3_, Fe(NO_3_)_3_, Fe(SCN)_3_) yielded complexes **1**–**3**. Dissolution of **1** together with either imidazole or 1-methylimidazole in n-butanol led to an exchange of the chlorido ligand in **1** (method b). The resulting complexes **5** and **6** precipitated from the reaction mixture [4]. Under basic conditions, the μ-oxo dimer **7** was formed from **1**. Acidification of **7** with acetic acid resulted in compound **4**.

#### 2.1.2. Characterization of the Complexes

Elemental analyses confirmed the elemental composition and the purity of the compounds. The purity of the complexes was further evaluated by high-performance liquid chromatography (HPLC) using an RP18 column with 35% methanol (MeOH)/65% phosphate-buffered solution (25 mM sodium dihydrogen phosphate; pH = 3 adjusted with phosphoric acid; flow rate: 1 mL/min) as eluent and a diode array detector (DAD) [20,21]. Complexes **1**–**6** were dissolved in MeOH (0.5 mM), and complex **7** was dissolved in ACN (0.5 mM). All compounds showed very similar retention times with a purity of at least 95% (Appendix A). This fact might be the consequence of an exchange of the axial ligand in the mobile phase. 

The lability of the ligand A was also visible by high-resolution mass spectrometry (HR-MS). The mass spectra were measured in the positive mode with an Orbitrap Elite mass spectrometer with electrospray ionization. In the spectra of all complexes (Appendix A), the cationic [[salophene]iron(III)]^+^ unit with a mass in the range of 370.0404–370.0421 (*m*/*z*) was detected. It resulted from the release of the axially positioned ligand (see Appendix A) during the measurement. In contrast to the investigations of Vančo et al. [4], the simple protonated form [imidazole[salophene]iron(III) + H]^+^ (at *m*/*z* 438.08) and [[[salophene]iron(III)]_2_ + CH_3_O]^+^ (at *m*/*z* 771.15) were not detected in the spectrum of **6**. The HR-MS spectrum of complex **7** (Appendix A) showed the [μ-oxo-[[salophene]iron(III)]_2_ + H]^+^ form at 757.0880 (*m*/*z*) and the adduct [μ-oxo-[[salophene]iron(III)]_2_ + [salophene]iron(III)]^+^ at 1126.1217 (*m*/*z*).

Fourier-transform infrared spectroscopy (FT-IR) revealed typical vibrations for the azomethine group ν(C=N), which occurred at 1598–1609 cm^−1^. Vibrations at 1299–1316 cm^−1^ could be assigned to the phenolate (ν(C_arom_–O)), which completed the chelates at the metal center [4].

Characteristic vibrations of the axially bound ligand in complexes **2**–**7** are listed in Table 1. Appendix A present the respective spectra. The frequencies of the NO_3_^−^, SCN^−^, and CH_3_-COO^−^ groups indicated a coordination to the metal [22,23,24,25]. In the case of the imidazole ligands, an increased number of C–H vibrations was visible. Furthermore, the spectra of **3** and **5** agree with those already described in the literature [4,23].

Somewhat more difficult was the assignment of **7** to the μ-oxo species, because characteristic vibrations could not unequivocally be assigned. However, the elemental analysis confirmed the dimer formation of two [salophene]iron(III) units. 

Oxo-bridged Fe(III) complexes show symmetric and asymmetric stretching vibrations of the Fe-O-Fe moiety with considerable variability in position and intensity. The greatest influence, however, resulted from the angle between oxygen and the iron centers [23,24].

Comparison of the spectra obtained with **1** (Appendix A) and **7** (Appendix A) documented for **7** in the range of 750 to 850 cm^−1^ (characteristic for the asymmetric Fe-O-Fe frequency) a strong and broad band at 749 cm^−1^, which we assigned to ν_as_(Fe-O-Fe). 

### 2.2. Biological Evaluation

The biological activity of the compounds was determined in the Cisplatin-resistant ovarian cancer cell line A2780cis and the mammary carcinoma cell line MDA-MB-231. As the complexes produced similar effects in both cell lines, only the results obtained from A2780cis cells are discussed and the data from MDA-MB-231 cells are submitted as Appendix A. Complexes **1**–**3** and **5**–**6** were stored as a 10 mM stock solution in DMSO. Cisplatin, **4**, and **7** were dissolved in DMF (10 mM). Cell-culture medium without fetal bovine serum (FBS) was used to adjust the final concentrations. 

#### 2.2.1. Effect on Proliferation and Metabolic Activity

The ability of the compounds to inhibit [^3^H]-thymidine incorporation into the DNA is an adequate parameter to quantify antiproliferative activity. Thus, the cells were incubated for 72 h with complexes **1**–**7** at concentrations of 0.1 µM, 0.5 µM, and 1 µM. Cisplatin was used at 1 µM and 5 µM. As expected, the effect of Cisplatin was low and the proliferation of the A2780cis cell line was only reduced at a concentration of 5 µM to 46% (Figure 1). Complexes **1**–**3** and **5**–**7** were 10-fold more effective than Cisplatin and decreased the proliferation regardless of the axial ligand used (proliferation at 1 µM: <10%; at 0.5 µM: 10–25%; at 0.1 µM: 60–72%; Figure 1). Only complex **4** was slightly less active (proliferation at 1 µM: 23%; at 0.5 µM: 47%; at 0.1 µM: 82%). 

To assess whether the antiproliferative effects were associated with cytotoxicity, a modified 3-(4,5-dimethylthiazol-2-yl)-2,5-diphenyltetrazolium bromide (MTT) assay was performed. This method quantifies the reduction of tetrazolium salts to colored formazan derivatives in the presence of functional mitochondria by photometric measurement. 

Cisplatin was inactive at 1 µM and 5 µM (metabolic activity at 1 µM: 101%; at 5 µM: 106%; Figure 2). Complexes **1**–**3** and **5**–**7** inhibited the metabolic activity about 20–25% less than the [^3^H]-thymidine incorporation (metabolic activity at 1 µM: 8–24%; at 0.5 µM: 24–44%; at 0.1 µM: >78%; Figure 2). Again, compound **4** induced the weakest effects and diminished metabolic activity to 60% at 1 µM, 77% at 0.5 µM, and 97% at 0.1 µM. 

All complexes were also investigated for their influence on the metabolic activity of the non-tumorous human stroma cell line HS-5 (Figure 3). At concentrations <1 µM, complexes **1**–**7** did not significantly reduce the viability of the cells, demonstrating tumor-specific effects at low concentrations.

#### 2.2.2. Cell-Death Induction

Decreased metabolic activity can drive cells into apoptosis, a mechanism of action already described in previous studies for the lead structure **1** in lymphoma and leukemia cells [8,16], as well as in ovarian adenocarcinoma cells [29].

Therefore, the activity of the effector caspases 3/7 in A2780cis cells after incubation with the complexes (1 µM) for 24 h was quantified by means of a luminescence assay. Compounds **1**, **4**, and **7** caused a 2-2.6-fold and compounds **2**, **3**, **5**, and **6** a 3-3.5-fold induction of caspases 3/7 activity (Appendix A). However, these values are too low to consider apoptosis as the preferred mode of cell-death for A2780cis cells. 

To gain a deeper insight into the mechanism of cell-death induction, complexes **1**, **2**, and **5** were selected and their effects on A2780cis cells were studied after incubation for 24 h by flow cytometry using Annexin V and propidium iodide (PI) staining to distinguish between viable, apoptotic, and dead cells. In the negative control, treated with the solvent DMSO only, the majority of the cells (79%) remained alive, 9% died, and 4% were apoptotic (Figure 4). As already indicated by the low activity of caspases 3/7, complexes **1**, **2**, and **5** only marginally caused apoptosis (8–20%; Figure 4). A cell population of 40–54% was identified as dead, induced by mechanisms other than apoptosis, e.g., ferroptosis or necroptosis.

In order to determine the kind of cell-death, A2780cis cells were simultaneously incubated for 24 h with the complexes as well as the ferroptosis inhibitor Ferrostatin-1 (Fer-1) or the necroptosis inhibitor Necrostatin-1 (Nec-1). Interestingly, Fer-1 alone (1 µM) caused 22% dead and 5% apoptotic cells; 68% of the cells were still alive (Figure 5). Fer-1 added concomitantly with **1** or **2** reduced cell death to the control level. The effect of **5** was only partially diminished by Fer-1. About 25% of the cells remained dead and a slight increase in apoptosis (15%) was determined. 

The addition of Nec-1 (20 µM) did not influence the cells compared to the solvent control (Figure 6). Incubation of Nec-1 together with the complexes only partially abolished the compounds’ inhibitory effects. The amount of living cells marginally increased from 17–32% (Figure 4) to 40% (Figure 6). These results clearly demonstrate that cell-death is mainly induced by ferroptosis with a minor involvement of necroptosis and limited apoptosis. 

#### 2.2.3. ROS Induction

Ferroptotic cells are characterized by smaller mitochondria and increased membrane density [30]. The induction of mROS is one of the modes of action of [salophene]iron(III) complexes [5]. Therefore, A2780cis cells were treated with compounds **1**, **2**, and **5** for 24 h and thereafter stained with reduced MitoTracker^®^Red-H_2_XROS to detect mROS. Furthermore, cell-membrane proteins were visualized with wheat germ lectin and Hoechst 33342 was used for imaging the nucleus. Even if live confocal microscopy did not allow quantification, the pictures clearly indicated enhanced mROS production in complex-treated cells compared to the control cells (Figure 7). Differences in mROS activity between the compounds, however, were not visible.

Quantifying ROS production is indirectly possible by simultaneous treatment with N-acetyl-L-cysteine (NAC), which acts as a scavenger of free radicals [31]. Analysis was performed by flow cytometry. NAC (1 mM) partially reduced the effects of **1**, **2**, and **5** so that 54–58% of living cells were detected, indicating that mROS was induced by the compounds (Figure 8).

To further investigate the influence of the compounds on mitochondria, the carbocyanine dye JC-1 was used as a ratiometric indicator of the mitochondrial membrane potential ΔΨm. JC-1 was added to A2780cis cells for 30 min after an incubation period of 24 h with 0.1 and 0.5 µM of the compounds. The dye accumulated in the mitochondria dependent on the membrane potential and was photometrically quantified. The complexes did not change the membrane potential (ΔΨm = 92–110%, Appendix A) at the low concentration of 0.1 µM, but they clearly diminished ΔΨm at 0.5 µM (Appendix A). While **1** (ΔΨm = 52.9%) and **2** (ΔΨm = 56.7%) showed the same effectivity, **5** caused a ΔΨm of 69.7%. All these results agree with the decreased metabolic activity (Figure 2) and underline the influence of [salophene]iron(III) compounds on mitochondria.

#### 2.2.4. Lipid Peroxidation

Ferroptosis is typically accompanied by accumulation of lipid peroxides resulting from ROS generated through the Fenton reaction. As the cytotoxic effects of [salophene]iron(III) complexes were suppressed by Fer-1, it was of interest to analyze lipid peroxidation as an indication of lipid ROS. Accumulation of lipid peroxides can be visualized with BODIPY™ 581/591 C11 dye. The oxidation of the phenylbutadiene residue of the fluorophore by these species shifted the fluorescence from red to green, which is analyzed by flow cytometry. 

A2780cis cells were treated with **1**, **2**, or **5** at concentrations of 0.1 µM and 0.5 µM for 8 h, 12 h, and 24 h, respectively, before adding the dye for 30 min. Independent of the incubation time, lipid peroxidation was low in each case at 0.1 µM (Table 2). At a concentration of 0.5 µM, however, 7.2% (**1**), 11.2% (**2**), and 7.7% (**5**) of the cells were positively tested for lipid ROS after 8 h. After 24 h, lipid peroxidation decreased to control levels.

### 2.3. Electrochemical Behavior of the Complexes

The above discussed results clearly demonstrate that [salophene]iron(III) complexes generate intracellular ROS, either as mROS or in a Fenton-like reaction. Both events caused cellular stress and led to cell death. However, no differences in biological activity were observed regardless of whether the complexes had an anionic or a neutral ligand. Therefore, the influence of the axial ligand on the redox potential of the complexes was investigated. The choice of solvent was of great importance, because complete exchange of the axial ligand in the mobile phase (MeOH 35%, phosphate buffer 65%, pH 3) has already been suggested by HPLC. Thus, aprotic solvents (DCM, ACN, DMF, and DMSO) were selected for cyclic voltammetry investigations.

DCM is a solvent without a tendency to coordinate to metals and allows the measurement of [salophene]iron(III) complexes dependent on the axially bound ligand. In contrast, DMSO is known to be a solvent with high affinity to metal complexes. It causes substitution reactions, as already demonstrated, e.g., in dichlorido platinum(II) complexes [18]. The binding trend of ACN and DMF is much lower and ligand exchange reactions are not preferred. 

Electrochemical measurements were performed on 1 mM solutions of the complexes in the respective anhydrous solvent with tetrabutylammonium hexafluorophosphate (TBAHPF, 0.5 M) as the supporting electrolyte. Before measurement, the solution was rinsed with argon to exclude the presence of oxygen. Standard potentials (E_1/2_) were calculated in relation to ferrocene (Fc, 2 mM) and are listed in Table 3. The corresponding voltammograms are shown in Appendix A. Exemplarily, the graphics of **1**–**3** in DMSO and DCM are depicted in Figure 9.

For complex **1**, E_1/2_ = −729 mV in DCM and E_1/2_ = −680 mV in ACN were determined. Complex **1** has already been part of various investigations regarding its redox behavior [13,14]. The difference of 49 mV between DCM and ACN agrees with that obtained by Carré and colleagues [13]. Compared to “free” iron(III) (E_1/2_ = −345 mV, Figure 10), a significant shift of 384 mV was observed, resulting from the electron delocalization from the salophene ligand to iron(II/III). Analysis of a mixture of **1** and FeCl_3_ in DMSO identified two well-separated redox pairs (Figure 10). This experiment further confirmed the stability of the [salophene]iron(III/II) moiety during the redox reaction without release of the central iron ion. In the voltammogram of **1** (Appendix A), free iron(III/II) is not visible. Consequently, intracellular Fenton-like reactions have to be ascribed to the chelated iron(III/II) species.

In the non-coordinative solvent DCM, the pentavalent complex was present, while ACN was attached as a further ligand at iron(III) without causing an ACN/Cl exchange. The octahedral arrangement resulted in a higher electron density at the metal and increased the standard potential. The substitution of the Cl^−^ ligand, as proposed by Carré et al., was proved by E_1/2_ = −728 mV of the resulting DMSO complex (Table 3). 

E_1/2_ of complexes **2** and **3** bearing electron-withdrawing ligands (NO_3_^−^ and SCN^−^) at the axial position were reduced in DCM to −502 mV and −572 mV, respectively (Table 3). The dissolution of **2** and **3** in ACN shifted E_1/2_ to −392 mV and −515 mV. In DMSO, the E_1/2_ = −723 mV and −724 mV pointed to a fast coordination of the solvent and a release of NO_3_^−^ and SCN^−^, which then acted as counter ions. 

Acetate represents an electron-rich ligand, increasing the electron density at iron(III). E_1/2_ of complex **4** decreased to −873 mV in DCM and −819 mV in ACN (Table 3). In DMSO, ligand exchange was observed again (E_1/2_ = −730 mV). 

Imidazoles are hard ligands for metal complexes. The imidazole–iron bond is stable and the initial complex can be measured in DCM and ACN. Complexes **5** and **6** displayed a comparable E_1/2_ of about −660 mV in both solvents. Dissolution in DMSO shifted E_1/2_ to −722 mV and 726 mV (Table 3). 

The μ-oxo complex **7** showed standard potentials of −1427 mV (DCM), −1376 mV (ACN), and −1403 mV (DMSO) comparable to that already described [14] and confirmed the existence of the oxygen bridged dimer. In DMSO, an additional species with E_1/2_ = −724 mV appeared. This potential points to the formation of a complex with a structure comparable to that built by other complexes in DMSO. These data document a higher stability of **7** upon coordination of O_2_^−^; however, a small amount of the monomeric DMSO species was formed from complex **7**. 

We further addressed the question of the complex stability in the presence of oxygen or water. Therefore, we investigated **1** (1 mM) dissolved in DMSO (3 mL) in the presence of water (0.03 mL). As depicted in Appendix A, the voltammogram does not show any additional peaks. Addition of a weak base such as 2-fluoropyridine caused the formation of small amounts of **7** (Appendix A). If a stronger base is used, the μ-oxo complex is quantitively formed in a simple acid–base reaction (1) (see experimental part).
2 chlorido[salophene]iron(III) + H_2_O + 2 Base → μ-oxo-[[salophene]iron(III)]_2_ + 2 BaseH^+^ + 2 Cl^−^(1)

In anhydrous solvents, the complexes are extremely sensitive to O_2_. If the complexes were dissolved in O_2_-free anhydrous DMSO, the [DMSO[salophene]iron(III)]^+^ complex was formed in each case, with E_1/2_ in the range of −720 to −730 mV (for **1**, Figure 11). As soon as traces of oxygen were present, the peak current of the monomeric complex strongly decreased and a new peak at about −1395 mV appeared, which was assigned to complex **7**. The reaction of the monomer to the μ-oxo-dimer was reversible, because rinsing the solution with argon during the measurement with several scans strongly decreased the peak current of **7** (Figure 11).

The results of this study clearly demonstrate that caution should be exercised when testing the [salophene]iron(III) complexes with variable axially bound ligands. If solvents with coordinative characteristics, such as DMSO or DMF, are used, a rapid exchange takes place and a consistent solvent adduct is formed.

Although obtained from different initial complexes, nearly identical in vitro results were received for **1**–**7**. For example, during the preparation of the stock solutions of **1** and **5**, the imidazole ligand and the chloride were replaced by DMSO. 

Therefore, the stability of compounds **1**–**7** in various stock solutions was investigated by cyclic voltammetry and HPLC. Only in the case of **7** a timely limited proof of the initial dimer was possible, with a slow degradation to monomers. The salophene ligand, however, is chelate-bound and is not labilized by solvent molecules [20].

One problem that has not yet been solved is the behavior of the complexes in aqueous solutions. Especially in cell-culture medium, reactions with present components and exchange of the axial ligand are possible.

In light of these data, the results of other publications have to be critically reconsidered. Complex **5** and related compounds with 1,2,4-triazol-1-ido, benzo[d][1,2,3]triazol-1-ido, 5-aminotetrazol-1-ido, 5-phenyltetrazol-1-ido, and 5-methyltetrazol-1-ido ligands were previously synthesized by Vančo et al. [4] and tested in different cell lines. Complex **5**, dissolved in DMF, exhibited strong antiproliferative effects on the A2780 ovarian cancer and the MCF-7 breast cancer cell lines (IC_50_ = 0.33 and 0.70 μM, respectively), very similar to the activity reported in this paper on A2780cis and MDA-MB-231 cells.

All of their ligand-modified derivatives (solvent DMF) also indicated activity in A2780 (IC_50_ = 0.06–0.35 μM) and MCF-7 cells (IC_50_ = 0.35–0.70 μM) nearly identical to **5**. Furthermore, in other tested cell lines, the complexes caused antiproliferative effects nearly independent of the used axial ligand. Therefore, it can be assumed that in these cases, a substitution of the ligand by DMF occurred at least partially and, thus, impaired the biological effects. Similar conclusions can be drawn for the methoxy-substituted chlorido[salophene]iron(III) complexes synthesized by Lange et al. [29], which were dissolved in DMSO. They also very likely tested the formed Fe(III)-DMSO derivative instead of the chlorido complex. The same holds true for our complexes, regardless of whether they were dissolved in DMSO or DMF [5,6,7,8,16,17].

## 3. Materials and Methods

### 3.1. Materials

Analytical data were obtained with the following instruments: 

^1^H-NMR spectroscopy: “Mars” 400 MHz Avance 4 Neo spectrometer (Bruker) at 400 MHz; solvent: DMSO-d6 with tetramethylsilane as internal standard. Chemical shifts are given in parts per million (ppm). Coupling constants are given in Hertz (Hz). 

FT-IR spectroscopy: Bruker ALPHA FT-IR-P spectrometer with 32 scans in a wavenumber range of 4000–400 cm^−1^ using a resolution of 4 cm^−1^. 

High-resolution mass spectrometry: Thermo Fisher Scientific (Waltham, MA, USA) Orbitrap Elite mass spectrometer. 

Elemental analysis: UNICUBE^®^—Elementar. 

High-performance liquid chromatography: Shimadzu Nexera-i-LC-2040C-3D using a Chromolith^®^ HighResolution RP-18 endcapped 100–4.6 mm column. The purity of all compounds was ≥95%. The pH value of the buffer solution for the HPLC was tested with an inoLab^®^ pH 7110 pH meter and was adjusted with a two-point calibration. 

Cyclic voltammetry: BioLogic SP-150 as a potentiostat. A conventional three-electrode cell with a platinum-wire counter-electrode, an Ag/AgCl-electrode with saturated NaCl solution as a pseudo-reference electrode, and a glassy carbon electrode as a working electrode were used. Ferrocene (2 mM) was applied as an internal standard. The supporting electrolyte Bu_4_NPF_6_ (TBAHPF) was applied as received. The EC-Lab V11.31 software was utilized. 

All reagents, solvents, and other chemicals were purchased from Sigma-Aldrich, TCI chemicals or Alfa Aesar. Solvents were distilled prior to use.

### 3.2. Synthesis of Compounds

#### 3.2.1. Synthesis of the *N,N′*-disalicylidene-1,2-phenylenediamine Ligand [19]

1,2-Phenylenediamine (4.43 g, 41.0 mmol) was dissolved in 200 mL EtOH and 2-hydroxybenzaldehyde (10.0 g, 82.0 mmol) was added. The mixture was refluxed at 78 °C for 2 h. Afterward, approximately half of the solvent was evaporated. The mixture was cooled to rt and stored overnight. The precipitate was filtered off and washed with 50 mL of EtOH and 50 mL of MeOH. The product was dried under reduced pressure. Yellow powder, yield 12.1 g (38.3 mmol, 93.4%). 

^1^H-NMR (DMSO-d_6_, 400 MHz), δ [ppm]: 12.94 (s, 2H, Ar-OH), 8.94 (s, 2H, =C-H), 7.67 (dd, 2H, Aryl-H, ^3^J = 6.25 Hz, ^4^J = 1,48 Hz), 7.49 – 7.40 (m, 6H, Aryl-H), 6.98 (ddd, 4H, Aryl-H, ^3^J = 6.25, ^4^J = 1.00 Hz). FT-IR (ν_max_ (cm^−1^)): 3052 w, 1609 m (C=N), 1559 m, 1478 m, 1274 m, 755 s.

#### 3.2.2. Synthesis of the Complexes **1–3** (Method a)

One eq. of *N,N′*-disalicylidene-1,2-phenylenediamine was dissolved in EtOH (0.2 mol/L) and 1 eq. of the respective iron(III) salt was added to the mixture and refluxed at 78 °C for 2 h. The mixture was cooled to rt and stored overnight. The precipitate was filtered off and washed with 10 mL of EtOH and 10 mL of MeOH. The product was dried under reduced pressure. 

Chlorido[salophene]iron(III) (**1**). From *N,N′*-disalicylidene-1,2-phenylenediamine (0.20 g, 0.63 mmol) and iron(III) chloride anhydrous (0.29 g, 1.8 mmol). Brown powder, yield 382 mg (0.94 mmol, 52.3%). FT-IR (ν_max_ (cm^−1^)): 3048 w, 1603 m (C=N), 1577 m, 1379 m (C–N), 1313 m (C–O), 1232 w, 759 m, 537 m. HR-MS (MeOH): *m*/*z* calculated for C_20_H_14_ClFeN_2_O_2_: 405.0093; found: 370.0404 [M-Cl^−^]^+^. Anal. Calcd**.** (%) C_20_H_14_ClFeN_2_O_2_ × 0.5 EtOH; C 58.84; H 4.00; N 6.54. Found (%) C_20_H_14_ClFeN_2_O_2_ × 0.5 EtOH; C 59.00; H 4.06; N 6.61.

Nitrato[salophene]iron(III) (**2**). From *N,N′*-disalicylidene-1,2-phenylenediamine (0.16 g, 0.49 mmol) and iron(III) nitrate nonahydrate (0.20 g, 0.49 mmol). Purification by precipitation from acetone. Black powder, yield 0.15 g, (0.49 mmol, 72.3%). FT-IR (ν_max_ (cm^−1^)): 1705 w (NO_2_), 1601 m (C=N), 1577 m, 1381 m (C–N), 1316 m (C–O), 1279 s (N=O), 1003 m, 979 m (N–O) 758 m, 537 m. HR-MS (MeOH): *m*/*z* calculated for C_20_H_14_FeN_3_O_5_: 432.0283; found: 370.0416 [M-NO_3_^−^]^+^. Anal. Calcd. (%) for C_20_H_14_FeN_3_O_5_ × 0.5 acetone × 0.5 H_2_O: C 54.92; H 3.86; N 8.94. Found (%) C 54.85; H 4.19; N 9.04.

Thiocyanato[salophene]iron(III) (**3**). From *N,N′*-disalicylidene-1,2-phenylenediamine (0.20 g, 0.62 mmol) and iron(III) thiocyanate (0.14 g, 0.62 mmol). Black powder, yield 0.21 g, (0.48 mmol, 79.1%). FT-IR (ν_max_ (cm^−1^)): 2061 m, 2036 s (SCN), 1601 m (C=N), 1576 m, 1532 s, 1378 m, 1316 m (C–O), 744 s, 537 m. HR-MS (MeOH): *m*/*z* calculated for C_21_H_14_FeN_3_O_2_S: 428.0156; found: 370.0404 [M-SCN^−^]^+^. Anal. Calcd. (%) for C_21_H_14_FeN_3_O_2_S × 0.5 MeOH: C 58.12; H 3.63; N 9.46. Found (%) C 58.09; H 3.49; N 9.66.

#### 3.2.3. Synthesis of Complex **4**

Acetato[salophene]iron(III) (**4**): Compound **7** (0.30 g, 0.4 mmol) was dissolved in 100 mL of DCM and acetic acid (1.14 mL, 20.0 mmol) was added to the mixture and refluxed at 40 °C for 1 h. The mixture was cooled to rt and the solvent was evaporated. The solid was washed with n-butanol and EtOH and dried under reduced pressure. Dark brown powder, yield 0.21 g, (0.49 mmol, 61.4%). FT-IR (ν_max_ (cm^−1^)): 1607 (C=N), 1578, 1531, 1525s (COO), 1481 m, 1460, 1433, 1382 (C–N), 1310 (C–O), 1189, 1147, 1031 m, 561 w, 513. HR-MS (MeOH): *m*/*z* calculated for C_22_H_17_FeN_2_O_4_: 429.0538; found: 370.0412 [M-CH_3_COO^−^]^+^. Anal. Calcd. (%) for C_22_H_17_FeN_2_O_4_ × 0.5 EtOH: C 61.08; H 4.46; N 6.19. Found (%): C 61.07; H 4.13; N 6.50.

#### 3.2.4. Synthesis of the Complexes **5–6** (Method b) 

One eq. of complex **1** was dissolved in 50 mL n-butanol and 8 eq. of the respective imidazole were added. The product precipitated from the reaction mixture. 

Imidazole[salophene]iron(III)] chloride (**5**). From **1** (0.36 g, 0.89 mmol) and imidazole (0.48 g, 7.1 mmol). Black powder, yield 0.23 g (0.48 mmol, 53.7%). FT-IR (ν_max_ (cm^−1^)): 3131 w (C–H), 3095 w, 1598 m (C=N), 1575 m, 1538 m, 1379 m (C–N), 1299 m (C–O), 1062 m, 1033 w, 969 w, 746,71 s, 656 w 537 m. HR-MS (MeOH): *m*/*z* calculated for C_23_H_18_FeN_4_O_2_: 438.0774 [M]^+^; found: 370.0410 [M-imidazole]^+^. Anal. Calcd. (%) for C_23_H_18_ClFeN_4_O_2_: C 58.32; H 3.83; N 11.83. Found (%):C 58.18; H 3.89; N 11.81. 

1-Methylimidazole[salophene]iron(III) chloride (**6**). From **1** (0.20 g, 0.49 mmol) and 1-methylimidazole (0.32 g, 3.9 mmol). Drying under reduced pressure after addition of 10 mL ACN. Black powder, yield 0.07 g (0.49 mmol, 29.2%). FT-IR (ν_max_ (cm^−1^)): 3126 w (C–H), 3109 w, 1600 m (C=N), 1576 m, 1533 m, 1378 m (C–N), 1308 m (C–O), 1027 w, 1033 w, 969 w 748 s, 659 w, 535 m. HR-MS (MeOH): *m*/*z* calculated for C_24_H_20_FeN_4_O_2_: 452.0930 [M]^+^; found: 370.0418 [M-1-methylimidazole]^+^. Anal. Calcd. (%) for C_24_H_20_ClFeN_4_O_2_ × 0.5 ACN × 0.5 H_2_O: C 58.05; H 4.38; N 12.19. Found (%): C 57.86; H 4.44; N 12.06.

#### 3.2.5. Synthesis of Complex **7**

Complex **1** (0.25 g, 0.62 mmol) was dissolved in 200 mL EtOH and 2 g of a 29% aqueous triethylamine solution, from which **7** precipitated as a red solid. The precipitate was washed with water, methanol, and EtOH and dried under reduced pressure. Yield 0.18g (0.24 mmol, 77.2%). FT-IR (ν_max_ (cm^−1^)): 1604 s (C=N), 1577 m, 1531 m, 1314 m (C–O), 805 m, 785 w, 749 s, 534 m. HR-MS (MeOH): *m*/*z* calculated for C_40_H_28_Fe_2_N_4_O_5_: 756.0759; found: 755.0880 [M+H^+^]^+^, 370.0421 [[salophene]iron(III)]^+^. Anal. Calcd. (%) for C_40_H_28_Fe_2_N_4_O_5_ × 1 EtOH: 62.94; H 4.15; N 6.99. Found (%): C 63.10; H 3.90; N 7.35.

### 3.3. Cyclic Voltammetry

#### 3.3.1. Procedure of Sample Preparation for Cyclic Voltammetry

Ferrocene (2 mM), the respective complex (1 mM), and the supporting electrolyte (TBAHPF, 0.5 M) were weighed into preheated volumetric flasks and solved in a dry solvent with the help of an ultrasound bath immediately before the measurements. About 3 mL of the solution was then transferred via syringe to a micro-cell. The solution was flushed with argon by inserting the argon line outlet for a few minutes. Thereafter, the argon line outlet was placed on top of the solution to form an argon atmosphere.

#### 3.3.2. Analysis and Normalization of the Voltammograms

Five scans (cycles) at a scan rate of 100 mV/s were performed for each measurement. Additional experiments without ferrocene were performed, to assure that ferrocene did not cover additional peaks. Calculation of the anodic and cathodic standard potentials were performed via EC-Lab. With the calculated anodic and cathodic peak potential of ferrocene, the *x*-axis (E vs. Fc) of the voltammograms were set to 0 V.

### 3.4. General Cell-Culture Methods

The ovarian carcinoma cell line A2780cis was kindly provided by the Department of Gynecology, Medical University Innsbruck, Innsbruck, Austria. The mammary carcinoma cell line MDA-MB-231 was purchased from DSMZ—German Collection of Microorganisms and Cell Cultures, Braunschweig, Germany. HS-5 cells were kindly provided from the Tyrolean Cancer Research Institute. All cell lines were cultured in RPMI 1640 without phenol red (BioWhittaker, Lonza, Walkersville, MD, USA) supplemented with fetal bovine serum (10% FBS, Biowest, Nuaillé, France), L-Glutamine (2 mM), Penicillin (100 U ml^−1^), and Streptomycin (100 µg ml^−1^) (all from Sigma-Aldrich, Vienna, Austria) at 37 °C under a 5% CO_2_/95% air atmosphere and were passaged twice weekly. To maintain Cisplatin resistance, the A2780cis cells were incubated every second week with Cisplatin at a concentration of 1 µM. Complexes **1**–**3** and **5**–**6** were dissolved in DMSO, and complexes **4** and **7** as well as Cisplatin were dissolved in DMF. Upon dilution with RPMI 1640 cell-culture medium, the final test concentration was reached. Fer-1, Nec-1, and NAC were purchased from Sigma-Aldrich.

#### 3.4.1. Analysis of Proliferation

Logarithmically growing cells were seeded in triplicates in flat-bottomed 96-well plates with a density of 1 × 10^4^ cells in 50 µL per well. After incubation overnight at 37 °C under a 5% CO_2_/95% air atmosphere, the compounds were added to achieve the test concentrations at a final volume of 150 µL. During the last 12–16 h of incubation, each well was exposed to [^3^H]-thymidine (2 Ci mmol^−1^, Hartmann Analytic, Braunschweig, Germany). Cells were harvested after a total incubation time of 72 h by a semiautomated device and [^3^H]-thymidine uptake into cells expressed as counts per minute (cpm) was measured in a scintillation counter (Microbeta Trilux, Perkin Elmer, Waltham, MA, USA). The proliferation in the absence of complexes (as a negative control) was set at 100% and the compound’s activity was calculated as the percentage of the control.

#### 3.4.2. Analysis of Metabolic Activity

Logarithmically growing cells were seeded and treated as described above (analysis of proliferation). After 72 h, the cells were examined for their metabolic activity using a modified MTT assay (EZ4U kit; Biomedica, Vienna, Austria) according to the manufacturer’s protocol. Unspecific staining generated by the FBS-containing medium was excluded by subtracting the optical density of the cell culture medium. Metabolic activity in the absence of the compounds was set at 100%.

#### 3.4.3. Analysis of Caspases 3/7 Activation

Logarithmically growing cells were seeded and treated as described above (analysis of proliferation). However, the supernatant was transferred already after 24 h into a white 96-well plate and the induction of apoptosis was recorded as caspases 3/7 activity using the Caspase-Glo 3/7 kit (Promega, Madison, WI, USA) according to the manufacturer’s recommendations. Values were normalized to the luminescence values of the medium control only. All values were calculated according to the caspases 3/7 activity of untreated cells, which was set at 1.

#### 3.4.4. Determination of Cell-Death and Indirect ROS Production by Flow Cytometry

Logarithmically growing cells were seeded in flat-bottomed 96-well plates with a density of 2 × 10^4^ cells in 50 µL per well. The cells were incubated at 37 °C under a 5% CO_2_/95% air atmosphere for 24 h with each substance at a concentration of 1 µM, without or in combination with either Fer-1 (1 µM), Nec-1 (20 µM), or NAC (1 mM). After 24 h, the cells were detached with Accutase and washed with 1× Annexin buffer. Thereafter, cells were double-stained in 50 µL of 1× Annexin buffer with 1 µL of Annexin V (MabTag GmbH, Friesoythe, Germany) conjugated to FITC dye (green fluorescence) and 1 µL of the red fluorescent dye propidium iodide (PI; Sigma-Aldrich), which allows discrimination between viable (Annexin V−/PI−), apoptotic (Annexin V+/PI−), and dead (Annexin V+/PI+) cells. After incubation for 15 min at 4 °C, in the dark flow, cytometric analysis was performed on a FACSCanto II (Becton Dickinson, San Jose, CA, USA).

#### 3.4.5. ROS Staining

Cells were seeded in duplicates with a density of 2 × 10^4^ cells in 50 µL per well on an 8-well Nunc™ Lab-Tek™ Chambered Coverglass dish (Thermo Fisher Scientific, Rochester, NY, USA) and incubated for 24 h together with the compounds. Immediately before analysis, 5 µL of 20 µM HEPES buffer (Biochrom GmbH, Berlin, Germany) was added to each cavity. Subsequently, 2 µL of reduced MitoTracker^®^Red-H_2_XROS (Invitrogen by Thermo Fisher Scientific, Eugene, OR, USA) and 2 µL of Wheat Germ Agglutinin (WGA)-Alexa Fluor™ 488 conjugate (Invitrogen by Thermo Fisher Scientific, Eugene, OR, USA) were added and incubated for 15 min. Hoechst 33342 (Invitrogen by Thermo Fisher Scientific, Eugene, OR, USA) was added 5 min before the cells were analyzed by confocal microscopy utilizing an inverted microscope (Zeiss Axio Observer Z1, Zeiss, Oberkochen, Germany) in arrangement with a spinning disk confocal system (UltraVIEW VoX, PerkinElmer, Waltham, MA, USA). All the images were generated using a 40× water immersion objective (Zeiss, Vienna, Austria).

#### 3.4.6. Determination of Mitochondrial Membrane Potential

The determination of the mitochondrial membrane potential was performed with the JC-1 mitochondrial membrane potential assay kit (Abcam plc., Cambridge, MA, USA) according to the manufacturer’s instructions. The cells were seeded in triplicates at a density of 1.5 × 10^4^ cells per 50 µL and incubated for 24 h with compounds **1**, **2**, and **5** (0.1 µM and 0.5 µM). Thirty minutes before the end of the incubation time, 100 µL of JC-1 dye was added to each well. Cells were washed twice with 100 µL of 1× dilution buffer. Analysis was performed at 535 nm/590 nm on the Enspire Multimode Plate Reader (Perkin Elmer).

#### 3.4.7. Lipid Peroxidation Staining with BODIPY™ 581/591 C11

Cells (2.5 × 10^5^) were seeded in a 12-well plate (Greiner Bio-One International GmbH, Kremsmünster, Austria) in 4 mL of medium. After incubation at 37 °C overnight under a 5% CO_2_/95% air atmosphere, the compounds were added at concentrations of 0.1 µM and 0.5 µM. After 8 h, 12 h, and 24 h, the cells were detached with Accutase and centrifuged at 200 rcf for 5 min. Meanwhile, a 2.5 µM BODIPY™ 581/591 C_11_ staining solution (Invitrogen by Thermo Fisher Scientific, Eugene, OR, USA) was prepared in phosphate-buffered saline (PBS) (Lonza, Verviers, Belgium) and the cells were resuspended in 100 µL of the staining solution and incubated for 30 min at 37 °C in the dark. After centrifugation for 10 min at 200 rcf and 4 °C, the pellet was resuspended in 200 µL of PBS and immediately analyzed by flow cytometry on the FACSCanto II.

## 4. Conclusions

The complexes inhibited proliferation and reduced metabolic activity in two cancer cell lines. Ferroptosis was identified to contribute to the mode of action accompanied by induction of lipid peroxidation and diminished mitochondrial membrane potential. However, identical biological activity was detected for almost all compounds. Only **4**, which was dissolved in DMF due to solubility reasons, was less effective. Therefore, the influence of the solvent on the ligand binding was investigated in more detail. Combining HPLC with cyclic voltammetry proved that the chelate at the [salophene]iron(III) complexes was stable, but the axial ligands were sensitive toward exchange reactions in strongly dissociative solvents. Measurements with traces of oxygen further showed that the complexes have a high affinity to oxygen and form a μ-oxo-bridged complex. Therefore, the nearly identical biological effects of complexes **1**–**3** and **5**–**6** might be the consequence of a transformation into the DMSO[salophene]iron(III) complex. 

## Data Availability

Not applicable.

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
