# Peer review of "Investigations on the Influence of the Axial Ligand in [Salophene]iron(III) Complexes on Biological Activity and Redox Behavior"

_ijms, 2023, doi:10.3390/ijms24032173_

Round 1

Reviewer 1 Report

It is an interesting and well-written manuscript that could be considered for publication after revision.

The stability of the complexes in solutions used for the biological studies should be explored and discussed.

Author Response

Thank you very much for your comment. Please see our response in the attachment.

Reviewer 2 Report

Descher et al. present a family of anionic and neutral [salophene]iron(III)) derivatives. The m-oxo dimer inhibited proliferation, reduced metabolic activity and increased mitochondrial reactive oxygen species (mROS). Electrochemical studies show that for all compounds a fast attachment of the solvent dimethyl sulfoxide and a release of the axial ligand, and in contrast, in DCM and ACN do not cause ligand exchange. The manuscript is present in adequate manner, and after minor revision, the draft should be suitable for publication in International Journal of Molecular Science.  

1.     The propose molecular formula for complexes 1 to 7 are only based on the elemental analysis, mass spectrometry and FTIR. All experimental data seems to corroborate the formula of the complexes. However, since the authors report the m/Z of each complex, the graphs of all the complexes measurements need be added since only the measurement of compound 1 is shown. For example, in the work of Vančo et al. (DOI: 10.1016/j.jinorgbio.2014.10.002) a detail study of the mass spectrometry measurements is presented.

2.     For compound 7, more experimental data should be given to confirm the existence of the m-oxo dimer, for example a more detail analysis of the FTIR spectra. Also, the authors should present the mass spectrometry measurements for this compound since they report a calculated m/Z value of 756.0759 and what they experimentally report is 370.0469 for a [M-Cl]-. More experimental data is needed to corroborate the molecular formula of compound 7.

3.     The authors cannot use the word “crystallized” to refer the solid they obtained. If the precipitate is not measure by X-ray diffraction (and observing peaks in the measurement, meaning that it is a crystalline material), they only can refer to the precipitate as a solid.

4.      As a suggestion, since the work of Vančo et al. (DOI: 10.1016/j.jinorgbio.2014.10.002) present quite similar systems a discussion could be done in the experimental section to improve the quality of the manuscript.  
